# A Case Report of Secondary Syphilis Co-Infected with Measles: A Diagnostic Dilemma with Fever and Rash

**DOI:** 10.3390/tropicalmed7050070

**Published:** 2022-05-09

**Authors:** Hisham Ahmed Imad, Ploi Lakanavisid, Phimphan Pisutsan, Kentaro Trerattanavong, Thundon Ngamprasertchai, Wasin Matsee, Watcharapong Piyaphanee, Pornsawan Leaungwutiwong, Wang Nguitragool, Emi E. Nakayama, Tatsuo Shioda

**Affiliations:** 1Mahidol Vivax Research Unit, Faculty of Tropical Medicine, Mahidol University, Bangkok 10400, Thailand; wang.ngu@mahidol.edu; 2Thai Travel Clinic, Hospital for Tropical Diseases, Faculty of Tropical Medicine, Mahidol University, Bangkok 10400, Thailand; ploi.la@go.buu.ac.th (P.L.); phimphan@thaitravelclinic.com (P.P.); thundon.ngm@mahidol.ac.th (T.N.); wasin.mat@mahidol.edu (W.M.); watcharapong.piy@mahidol.ac.th (W.P.); 3Center for Infectious Disease Education and Research, Department of Viral Infections, Research Institute for Microbial Diseases, Osaka University, Suita 565-0871, Osaka, Japan; emien@biken.osaka-u.ac.jp (E.E.N.); shioda@biken.osaka-u.ac.jp (T.S.); 4Department of Preventive Medicine and Family Medicine, Faculty of Medicine, Burapha University, Chonburi 20130, Thailand; 5Department of Clinical Tropical Medicine, Faculty of Tropical Medicine, Mahidol University, Bangkok 10400, Thailand; 6Hull York Medical School, University Road, Heslington, York, YO10 5DD, UK; hykt2@hyms.ac.uk; 7Tropical Medicine Diagnostic Reference Laboratory, Faculty of Tropical Medicine, Mahidol University, Bangkok 10400, Thailand; pornsawan.lea@mahidol.ac.th; 8Department of Microbiology and Immunology, Faculty of Tropical Medicine, Mahidol University, Bangkok 10400, Thailand; 9Department of Molecular Tropical Medicine and Genetics, Faculty of Tropical Medicine, Mahidol University, Bangkok 10400, Thailand

**Keywords:** measles virus, exanthema, *Treponema pallidum*, secondary syphilis

## Abstract

Fever and rash as manifestations of infection by microorganisms are collectively known as febrile exanthem. Since viruses are more frequently associated with fever and rash, these symptoms are thus impetuously termed viral exanthem. However, bacteria represent a frequently overlooked infectious etiology causing rash in humans. In addition, certain microbes may exhibit pathognomonic features that erupt during illness and facilitate clinical diagnosis. Conversely, coinfections often obscure the clinical characteristics of the primary disease and further challenge clinicians attempting to reach a diagnosis. We retrospectively looked at de-identified clinical data of a patient who presented to the Hospital for Tropical Diseases in Bangkok in July 2019 with complaints of fever and rash. The case involved a 35-year-old who presented with a 3-day history of fever, respiratory symptoms, myalgia, conjunctivitis, diarrhea, and a generalized maculopapular rash. On examination, the patient was febrile, tachycardic, and tachypneic, with a mean arterial pressure of 95 mmHg. A differential white blood cell count showed: leukocytes, 5800/µL; neutrophils, 4408/µL; lymphocytes, 406/µL; and platelets, 155,000/µL. Striking findings involving the integumentary system included Koplik’s spots and generalized maculopapular rash. Further serology revealed positive immunoglobulin (Ig)M and IgG for both measles and rubella virus, including reactive serology for *Treponema pallidum*. Here we describe the clinical course and management of this patient.

## 1. Introduction

Rash is a frequent puzzling manifestation encountered in primary-care clinical practice. The word “rash” commonly refers to any cutaneous eruption appearing over skin surfaces either acutely or chronically. When evaluating a patient who presents with a rash, the clinician considers whether the rash is life-threatening, highly transmissible, or benign [1]. A myriad of pathogens may cause the appearance of rash during febrile illness, and reaching a correct diagnosis is often challenging [2,3,4]. Some of these pathogens cause diseases identified from the earliest days of civilization, such as measles and syphilis, which have recently shown a trend toward re-emergence in modern communities [5,6].

The measles virus also known as the rubeola virus, is a single-stranded RNA morbillivirus in the *Param**yx**oviridae* family and was differentiated from smallpox by the Persian physician Rhazes during the Islamic Golden Age [7]. This highly infectious virus consists of pleomorphic virions. When inhaled into the body or transmitted by mucosal contact, an acute febrile exanthematous illness arises after an incubation period of approximately 10 days. Humans are the only reservoir for the virus [8].

The pathogenesis of measles begins within cells in the upper respiratory tract and is amplified within the lymphoid tissues, with the virus then disseminating to target organs, including the skin [9]. The disease is characterized by the abrupt onset of high-grade fever in association with the symptom triad of coryza, conjunctivitis, and cough, observed during a prodromal phase lasting 1–4 days [10]. The illness is more profound when the exanthemic cutaneous eruptions initially appear on the face and spread to the rest of the body in a centrifugal pattern that further evolves through coalescence and desquamation towards convalescence. Simultaneous with fever, the enanthemic mucocutaneous lesions (Koplik’s spots) pathognomonic to measles transiently erupt on the buccal mucosa opposite the first two molars [11,12]. Koplik described these spots as bluish-white specks on small, irregular spots of bright red color on the buccal mucosa membrane [12]. Involvement of the palms and soles in measles is rare but involvement of the palms has been reported to appear transiently overlapping the prodromal phase and acute illness [13].

Measles is almost always a self-limiting illness in children, with the rare occurrence of transitory post-measles immunosuppression through elimination of the existing antibody repertoire against other pathogens and long-term immunological sequelae [14,15,16]. Fatalities occur with complications involving the respiratory and central nervous system, and the risk of severe disease increases through adolescence to adulthood, with the illness atypically manifesting among immunocompromised individuals [16,17,18,19,20,21,22]. The diagnosis can be confirmed with molecular techniques, such as reverse transcription polymerase chain reaction or through serology, with measles virus-specific immunoglobulin (Ig)M antibody peaks seen when the rash is almost confluent, and the peak of IgG antibody observed with the resolution of symptoms, at which point the patient is no longer considered infectious. In Thailand, in 2019, a total of 9255 suspected cases of measles were reported to the Ministry of Public Health, with laboratory confirmation in 39%. That same year, 14 cases of measles were diagnosed at the Hospital for Tropical Diseases.

Another primeval pathogen that may induce florid skin lesions is *Treponema pallidum*, a spirochete that causes syphilis, historically identified as the Great Pox [23]. Similar to measles, humans are the only reservoir of this helix-shaped bacterium, and the route of transmission is by sexual contact, congenital acquisition, or by blood exposure [24]. The incubation period can be as short as 3 days or up to 3 months from the time of exposure [25]. This bacterium is considered infective during the early stages when spirochetemia appears. The primary stage consists of the eruption of a chancre either in the genitoperineal region or within the oral cavity. The syphilitic chancre is a painless, indurated ulcer that often goes unnoticed. The disease, left untreated, results in the seeding of bacterium (spirochete) within the body during the secondary stage, also known as “roseola syphilitica”, and the cutaneous manifestation appearing during this stage may mimic other diseases [26].

Although spirochetes may be visualized under dark-field microscopy of specimens collected from the primary syphilitic chancre, in addition to utilization of other, more sensitive molecular techniques such as nucleic acid amplification to detect *T. pallidum* genes, serology remains the technique most commonly used in clinical practice for the diagnosis of secondary, latent or tertiary syphilis, involving non-treponemal and treponemal tests; the former are utilized for screening, the latter for confirmatory diagnosis [27,28,29].

In secondary syphilis, the spectrum of polymorphic cutaneous eruptions is secondary to local host responses from the disseminated treponemes within host cells [26,30]. During this stage, lymphadenopathy is generalized and firm, non-tender without fluctuation and may persist for several weeks [31]. The indolent eruptions are initially transitory macular rashes that are elliptical in morphology, sparing involvement of the face [32]. Conversely, the rash that appears during late secondary syphilis shows a predilection for involving the face, including the forehead, and persists longer [33]. However, these slightly larger papules are flattened and sparsely scattered. Other distinct features of the rash observed during secondary syphilis include the symmetrical distribution of a rash that is pink or dusky-red in color, particularly involving the palms and soles [32,34].

The global incidence of both these preventable infectious diseases has been increasing over recent decades [35,36]. Measles outbreaks have become more frequent among communities with higher rates of rebuffing vaccination, leading to suboptimal vaccination coverage. With the advent of the acquired immunodeficiency syndrome (AIDS), epidemic syphilis cases grew in numbers among homosexual and bisexual communities [37,38]. In this high-risk group for sexually transmitted pathogens, co-infection with the human immunodeficiency virus (HIV) and syphilis are common and atypical manifestations masking the classical findings of syphilis have been reported [6,39,40,41,42]. In addition, co-infection of measles virus and the spirochete *T. pallidum* have not previously been reported.

The clinical presentation and in-hospital management of a patient diagnosed with both measles and syphilis is presented.

## 2. Materials and Methods

The patient described in this report presented to the Hospital for Tropical Diseases in Bangkok in 2019. De-identified data from the medical charts were retrospectively reviewed to extract clinical and laboratory data from the time of presentation until discharge from the hospital. Serological assays were used for confirmation and exclusion of pathogens. These included measles and rubella IgM and IgG by enzyme-linked immunoassay (ELISA) (Euroimmun, Lubeck, Germany), dengue non-structural protein 1 (NS1) antigen (Biosynex, Swiss S.A, Fribourg, Switzerland), hepatitis B surface (HBs) antigen and anti-hepatitis HBs antibody by electrochemiluminescence immunoassay or ECLIA (Roche, Mannheim, Germany), anti-HCV antibody (hepatitis C virus) (Humasis, Gyeonggi-do, Korea), HIV antibody/antigen by ECLIA (Roche, Mannheim, Germany), plasma rapid regain or RPR (Human, Weisbaden, Germany), and *T. pallidum* hemagglutination or TPHA (Rapid Labs, Colchester, UK).

## 3. Case Report

A case of acute febrile illness associated with rash in a patient is presented. A Thai man in his early thirties presented to the Fever Clinic at the Hospital for Tropical Diseases in Bangkok, Thailand. The patient was originally from a town a few hundred kilometers from the capital city. Being unemployed, he claimed to have been dwelling in Bangkok over the past several months.

The patient denied having any underlying medical history of chronic disease, any recent travel history or exposure to forested areas, or contact with animals or other sick individuals. Further, he had no apparent prior vaccination history. He had large tattoos covering the left front side of his body, which extended completely, covering the entire back, including an additional tattoo over the anterior aspect of the left lower extremity. He also claimed to have never received or donated blood for transfusion or had any intravenous drug use. In addition, he denied having recently consumed any over-the-counter medications or antibiotics. His sexual orientation was bisexual and he had a history of engaging in condomless sex with multiple partners, including those living with HIV with unknown viral load suppression status. The patient recalled that the approximate date of his last occasion of sexual intercourse was a month or two earlier, with a female partner.

He presented on the third day of illness, complaining of high-grade fever and coryza for the past 3 days. He also reported having developed red eyes and a rash over the face and neck from the second day of fever. Other signs and symptoms added further to the history of presenting illness, including loss of appetite and nausea, with multiple episodes of diarrhea. He had no complaints of vomiting or abdominal pain.

On examination, the patient was conscious and appeared clinically unwell, cachectic, and dehydrated with bilateral conjunctivitis and florid generalized symmetrical erythematous rash covering almost the entire body surface area, as seen in Figure 1.

Physical examination showed no stiffness of the neck to passive flexion or other signs suggestive of meningeal irritation. Both conjunctivae appeared to be injected without any purulent discharge. Visual acuity remained intact. Vital signs recorded at triage demonstrated an elevated body temperature of 40.5 °C, with increases in heart rate (120 beats/min) and mean arterial blood pressure (95 mmHg). The respiratory rate was rapid (34 breaths/min) with 97% oxygen saturation in room air.

On further observation during the physical examination, Koplik’s spots were present on bilateral buccal mucosa, which had erupted opposite the first two molars. Other findings within the oral cavity included an injected erythematous posterior pharyngeal wall. The tonsils showed no exudative deposits, and no other signs of severe immune compromise were apparent, such as oral candidiasis or features of hairy leukoplakia within the oral cavity. The cardiovascular system examination was normal, with typical heart sounds and no murmurs except for the tachycardia. Likewise, apart from the tachypnea, examination of the respiratory system yielded normal results with audible vesicular breath sounds and equal air entry in both lung fields. Findings for the central nervous system were unremarkable, with no motor or sensory deficits and a full Glasgow Coma Scale score of 15/15. The abdomen was soft and non-tender to palpation, with no evidence of hepato-splenomegaly. A florid generalized maculopapular rash was observed to be distributed symmetrically, involving the trunk, back, and periphery, which was described as non-pruritic and suggestive of a centrifugal distribution pattern. Suspicious rashes involving the palm and soles were identified, as shown in Figure 2.

Under a clinical diagnosis of measles, the patient was promptly and securely transferred and admitted to a negative-pressure room attached via an anteroom to disrupt and prevent aerosolized and droplet transmission. Intravenous fluids were administered to correct any dehydration, including other supportive medications, such as antipyretics, anti-histamines, and an anti-motility agent, including a cough suppressant. Blood and stool samples were collected and sent for culture studies, in addition to the routine hematological and biochemical investigations, including confirmatory serology, and for screening of suspected etiologies among the differential diagnoses and co-infections. Empirically, 2 g of ceftriaxone was added to the treatment. Laboratory investigation results at the time of presentation and during hospitalization are shown in Table 1.

The fever subsided 40 h after presentation to the hospital and vitals returned to normal ranges, as shown in Appendix A. On day 5 of illness, the patient showed marked clinical recovery, and was generally feeling much better. As no pending cultures had yielded bacterial growth, the antibiotic was discontinued. The rash over the body was no longer erythematous and only showed residual hyperpigmentation of the macules, with total clearing of palmar and plantar lesions. The patient was discharged from hospital on day 7 after the onset of symptoms with clinical recovery from measles and with a reactive RPR (titer of 1:16). A follow-up appointment with a repeated RPR to start treatment with benzathine benzylpenicillin G was advised.

## 4. Discussion

The Hospital for Tropical Diseases is a modern-day center of excellence in tropical medicine established in Bangkok on 23 February 1961. The hospital runs numerous specialized clinics, such as the Fever Clinic, which functions round the clock, 7 days a week, as a public service [43]. Patients diagnosed with tropical diseases are frequently referred from other hospitals and individuals suffering from acute febrile illness may opt to walk into the clinic or present as described in this report [44,45].

When formulating differential diagnoses for cases such as the one presented here, social lifestyles that pose various risks of pathogen transmission need to be considered. An important consideration would be the potential exposure to HIV, as acute HIV seroconversion may exhibit similar clinical manifestations to the present case, regardless of the known date of last exposure, in view of the prolonged incubation period [46,47]. Relatedly, antibodies against other blood-borne viruses, such as hepatitis B virus, were detected. In Thailand, vaccination against the hepatitis B virus was included in the nationwide expansion of the immunization program in 1992 [48]. We therefore postulate that the patient in the present case may not have received vaccination against hepatitis B virus. The detection of existing antibodies may be from an acquired infection in the past through exposure from tattooing or unsafe sexual practices with multiple partners. However, we could not further demonstrate any hepatitis B core antibody to reflect on the disease state during this admission.

A vaccine in the form of a live attenuated combination of measles, mumps, and rubella was introduced in Thailand under the expanded immunization program in 2008. Based on this information, we considered that the patient in this case may have been naïve to these viruses. Hence, other viral etiologies, such as rubella and parvovirus B19 infections, may also manifest with similar clinical findings to the presented case [49,50]. However, the classical manifestation of rubella in adults includes arthralgias, similar to the symptoms caused by arthritogenic viral infections [44,51,52,53]. Similarly, parvovirus B19 infection in adults may manifest with arthralgia, including enanthemic eruptions of Koplik’s spots [54]. These prodromal enanthemic eruptions are considered pathognomonic to measles but may occur with other viral infections. Koplik’s spots were present in 717 of 3,023 cases of suspected measles [55]. While a third of the cases in the same cohort had Koplik’s spots from measles, less than a quarter were caused by rubella, and a little over 5% were caused by other viruses.

The characteristic clinical manifestation with the peculiar extent of mucocutaneous rash, which followed an evolutionary course along with the clinical recovery, was suggestive of measles and less likely to be associated with other viral etiologies in the present case. Nevertheless, we were unable to perform further analyses to clearly exclude the possibility of viral co-infections.

Among the dreaded complications of measles are pneumonitis and bacterial pneumonia secondary to transient immune suppression or increased host vulnerability to other pathogens consequent on the elimination of the antibody repertoire observed in immune amnesia [14,56,57,58]. Similarly, lymphopenia in the present case might reflect the dysregulated cellular immune response in measles. In addition, to explain why the present case appeared tachypneic at triage but maintained normal oxygen saturation without supplemental oxygenation, we presumed that both tachypnea and tachycardia were due to a normal compensatory physiological response to the persistent elevated body temperature, despite the lack of a chest X-ray to clearly exclude the presence of lung infiltrates [59].

Other complications such as diarrhea have been reported to occur in 11% of patients with measles [60]. The present case also suffered several episodes of diarrhea, which may have contributed to dehydration and depleted electrolyte levels. However, the exact mechanisms of diarrhea in measles have not been fully elucidated. Previously, the presence of Koplik’s spots in the colon and mucosal inflammatory changes and deposition of giant cells within the visceral tissue, including other gastrointestinal complications such as acute appendicitis and cholecystitis, have been described in cases of measles [61,62,63].

Further, a previous study demonstrated that serological assays for rubella performed less accurately in the sense that false-positive findings resulted from cross-reaction with other viral infections or were associated with underperformance, with increased circulating serum proteins such as rheumatoid factor [64,65]. Hence, we advise clinicians to interpret serological results with caution. Furthermore, a previous report observed seroconversion in rubella 3–4 days after rash onset [66]. Retrospectively, in the present case, serum samples for serological investigations were collected on the third day of illness, representing the second day after the onset of rash. We therefore considered that the serological results for rubella may have been false positives in the present case. Additionally, our observations at the hospital and the epidemiological data for measles in Thailand suggest that measles is more common compared to rubella. Furthermore, other hyper-inflammatory states similar to Kawasaki disease may also show similar manifestations to the present case [67].

Some Gram-negative bacteria can cause similar symptoms, as observed during the early stages of meningococcemia before the development of purpura fulminans or fatal shock, or the rash observed in the spotted fever group of rickettsioses [68,69,70,71]. However, these etiological microbes may be the least likely to have caused disease in the present case, for the following reasons. First, the patient showed an absence of travel to known hyper-endemic regions for *Neisseria meningitis*, such as the African meningitis belt. Second, the hematological profile of the present case was more suggestive of a viral etiology than a bacterial one, as leukocyte counts remained within the lower quartiles of the normal ranges. The evidence of lymphopenia in addition to the absence of neutrophilia and leukocytosis were additional hematological indices that helped to exclude bacteremia in the present case. Nevertheless, *Rickettsia* species are intra-cytosolic bacteria that do not cause leukocytosis as commonly seen with most bacteremias or systemic bacterial infections. In addition, the patient had no recent history of contact with animals (dogs). Hence, the possibility of rickettsial infections endemic to Thailand, such as Thai tick typhus, was excluded as an etiology in the present case [72].

As expected, both treponemal and non-treponemal tests were reactive. False-positive treponemal serological assays have been reported in association with viral infection and during pregnancy, and false-negative results can occur due to the prozone phenomenon that is seen in the presence of an overabundance of antibodies against the spirochete [73,74]. Nonetheless, the patient had a substantial history of exposure to sexually transmitted diseases on multiple occasions. However, physical examination did not reveal any generalized lymphadenopathy or evidence of patchy alopecia, described as a “moth-eaten” pattern, another manifestation of secondary syphilis [75]. We think there may have been an overlap in cutaneous manifestations observed during secondary syphilis. Unfortunately, we did not have the opportunity to examine the genitalia or perianal region during the present hospitalization, which could have provided further evidence of syphilis based on the presence of condyloma lata, skin tag-like growths that arise from mucosal patches in secondary syphilis [76].

Interestingly, macules over the palms and soles rapidly resolved with no residual scarring. We therefore cannot precisely say whether empirical antibiotics administered on admission were effective in terms of bactericidal actions against the treponemes, as previously described [77]. Nevertheless, weekly intramuscular injections with benzathine benzylpenicillin G were planned for three consecutive weeks to achieve radical cure and prevent the development of tertiary syphilis. Penicillin remains the drug of choice to treat syphilis, and doxycycline has been described as an acceptable alternate option to treat early and late latent syphilis if penicillin cannot be used [78]. Also worth mentioning is the Jarisch Herxheimer reaction, which may transiently make the patient more symptomatic after initiating antibiotic treatment [79,80]. Although the clinical manifestations and serological results were compatible with a co-infection of measles and secondary syphilis, our observations were limited due to not having performed further additional tests such as PCR for virus detection. 

As alluded to in the Introduction, humans are the only reservoirs for both measles virus and *Treponema pallidum*. Therefore, theoretically, disruption of the transmission cycle through screening, vaccination, and treatment could achieve the eradication of these two diseases. In conclusion, viral illness brought the patient in the present case to the hospital, leading to the unmasking of a co-infection. A comprehensive history of the presenting illness, including past immunization, travel, and exposure history, can help narrow down the differential diagnoses of patients presenting with fever and rash.

## Figures and Tables

**Figure 1 tropicalmed-07-00070-f001:**
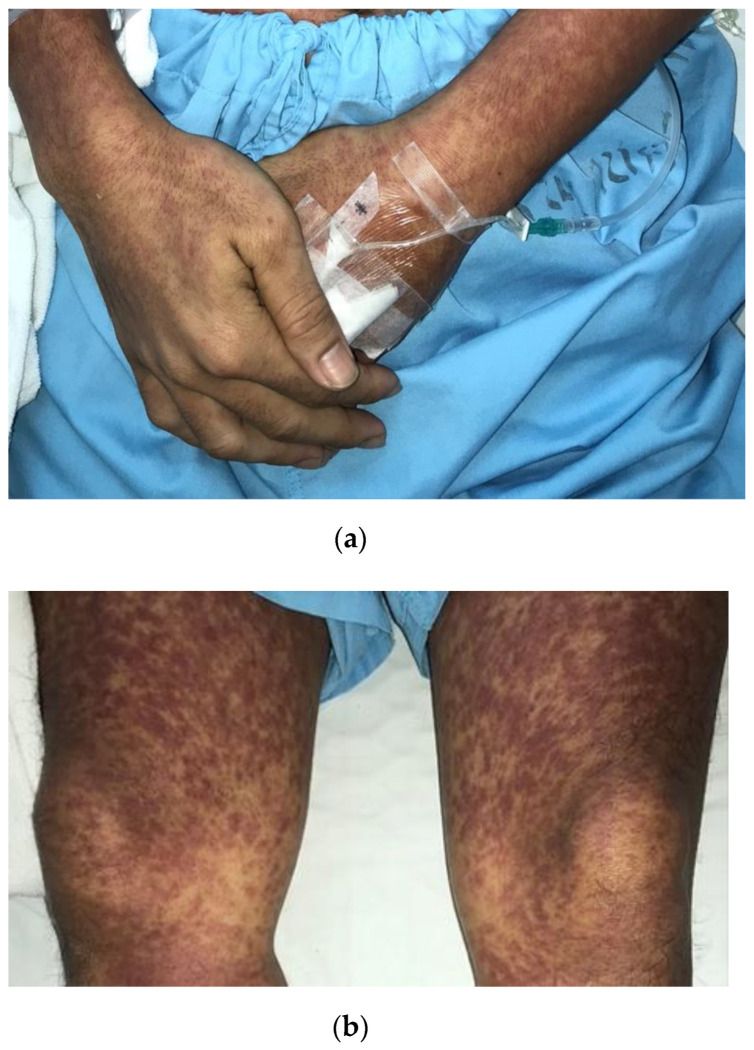
Florid maculopapular rash from measles (**a**,**b**). Confluent erythematous maculopapular rash of varying sizes symmetrically scattered over the trunk, with slight central clearing, showing extension to the (**a**) forearms (including the dorsal aspect of the palms) and (**b**) legs.

**Figure 2 tropicalmed-07-00070-f002:**
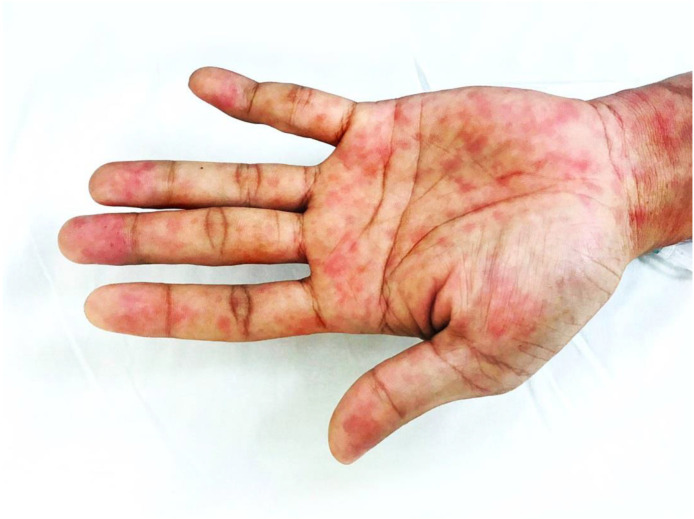
Non-blanching, erythematous macular rash 0.1–2 cm in diameter on the palm and confluent florid maculopapular rash visible on the anterior aspect of the wrist, initially extended inferiorly from the face, trunk, and extremities.

**Table 1 tropicalmed-07-00070-t001:** Laboratory parameters investigated during hospitalization.

Day of Illness	Third Day	Fourth Day
Hemoglobin g/dL (14.0–18.0)	16.8	16.7
Hematocrit % (40.0–54.0)	49.2	49.6
Leukocytes/µL (5000–1000)	5800	5100
Neutrophils/µL (2500–6000)	4408	3723
Lymphocytes/µL (1000–4800)	406	561
Monocytes/µL (200–800)	0	102
Eosinophils/µL (30–350)	0	0
Basophils/µL (0–300)	0	0
Band form/µL (0–4)	928	510
Atypical lymphocytes/µL (0–600)	58	204
Platelets/µL (150,000–450,000)	155,000	136,000
Direct bilirubin mg/dL (0.0–0.3)	0.1	
Total bilirubin mg/dL (0.0–1.2)	0.3	
Total protein g/dL (6.6–8.7)	7.1	
Albumin g/dL (3.5–5.2)	3.9	
Globulin g/dL (2.5–3.5)	3.2	
Alkaline phosphatase IU/L (40–129)	80	
Aspartate aminotransferase IU/L (0–40)	44	
Alanine transferase IU/L (0–41)	42	
Creatinine mg/dL (0.67–1.17)	1.05	
Blood urea nitrogen mg/dL (6–20)	15.4	
Sodium mmol/L (136–145)	130	
Potassium mmol/L (3.5–5.1)	3.6	
Chloride mmol/L (98–107)	96	
Dengue NS1	Negative	
Measles IgM	Positive	
Measles IgG	Positive	
Rubella IgM	Positive	
Rubella IgG	Positive	
Hepatitis B surface Ag	Negative	
Anti-hepatitis B surface Ab	Positive	
Anti-hepatitis C Ab	Negative	
Anti-HIV Ag/Ab	Negative	
RPR	Reactive	
TPHA	Reactive	
Blood culture	No growth	
Stool culture	No growth	

NS1: non-structural protein 1; IgM: immunoglobulin M; IgG: immunoglobulin G; Ag: antigen; Ab: antibody; RPR: rapid plasma reagin; TPHA: *Treponema pallidum* hemagglutination assay.

## Data Availability

The data presented in this study are available on request from the corresponding author. The data are not publicly available to ensure the privacy of the study participant.

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
