# Peer review of "A Case Report of Secondary Syphilis Co-Infected with Measles: A Diagnostic Dilemma with Fever and Rash"

_tropicalmed, 2022, doi:10.3390/tropicalmed7050070_

Round 1

Reviewer 1 Report

Overall comment:

Although expansive, it is a primer on the presentation, pathogenesis and anticipated lab results for a number of infectious etiologies causing a rash under consideration in this case.

There is just one line at the end of the abstract that proposes this to be a dual measles & syphilis infection, easily missed and should be reiterated at the end or beginning of the Discussion or Case report if that is the theme.

Comments & questions

Table 1 – should include normal ranges

1> You found an IgM positive response to both measles & rubella – could this not be a false-pos result ?

2> In acute cases of measles, especially in the first 3 days after rash onset, the IgM response and more often the IgG response is equivocal or absent. In contrast for rubella both the rubella IgM & IgG are present at rash onset. Why not take another blood at the time of discharge to record a seroconversion to one or both which would firm up the serologic diagnosis ? PCR for measles, if available, would also be another option to confirm the measles diagnosis.

3> Was there no titre provided for the positive RPR finding as the titre can be used as a guide to determine if it is early, late latent etc ? The TPPA is a confirmatory assay and remains positive despite treatment?

4> The fairly brisk resolution of the rash following his ceftriaxone is suggestive of a bacterial etiology vs virus so why not presumptively commence treatment for secondary syphilis given his risk behaviours and probability he may not go the full course ?

Author Response

>>Thank you for the comments, and we welcome all the suggestions. In the revised manuscript, we have changed the title for better clarity from “The first disease and the great imitator: a diagnostic dilemma with ancient maladies” to “A case report of secondary syphilis co-infected with measles: a diagnostic dilemma with fever and rash.”

>>We have included the normal ranges of the hematological and biochemical profile in Table 1, in the revised manuscript.

>> Yes, we considered the possibility of a false-positive result in the serology, and included a sentence in the revised manuscript (Line 294 -295)advocating for careful interpretation of the serological results. We ruled in measles after considering several clinical findings and the course of illness. Firstly, the presented case had the triad of conjunctivitis, coryza, and cough, infamously described in the literature about measles. The appearance and distribution of rash were consistent with measles, including the eruption of Koplik’s spots. Adults with rubella exhibit arthralgia and with the absence of arthralgia in the presented case suggest the unlikelihood of rubella. Additionally, we considered rubella to be epidemiologically far less common in Thailand compared to measles. To elaborate on this we have added the following sentences in the revised manuscript.

“(Lines 300-301) Additionally, the epidemiology of measles in Thailand is far more common compared to rubella.”

>> We appreciate this suggestion of utilizing PCR for diagnosis, and we have included the following sentence and have expressed the limitation of PCR for measles, in the revised manuscript.

“(Line 341-344) Although, the clinical manifestations and serology, were compatible with a co-infection of measles and secondary syphilis. We were limited not to have performed further additional tests like PCR for virus detection.”

>>In the revised manuscript, we have included the RPR titer and included an additional explanation of the diagnoses at discharge.

“(Lines 226-229) The patient was discharged from the hospital on day 7 after the onset of symptoms and following the clinical recovery from measles. Further,  as the investigation revealed a reactive RPR (titer of 1:16), a follow-up appointment to repeat the RPR and to start treatment with benzathine benzylpenicillin G was advised.

>>In the presented case, ceftriaxone was empirically started and discontinued cultures yielding no growth. Given the history provided by the patient, a serological workup was performed during this admission and additional workup and treatment were planned. In the revised manuscript, we have included the following sentence and provided the reference to the readers regarding the recommended therapeutic options in the treatment of syphilis.

(Lines 337-339) Penicillin remains the drug of choice to treat syphilis, and doxycycline has been described to be an acceptable alternative option to treat early and late latent syphilis, if penicillin cannot be used.

Reviewer 2 Report

Please see the comments attached. 

Author Response

>>Thank you for taking the time in reviewing our manuscript. We appreciate and welcome the comments and suggestions.

We appreciate the comment by the reviewer, regarding the interpretation of the serology, and the implication in diagnosis and associated seroconversion in clinical practice to correlate with the finding presented in this Case Report and have revised the manuscript to address this. Further, as suggested, we also have described the laboratory diagnosis of syphilis and its interpretation in the Introduction and have associated the findings in the Discussion.

the lines 85 to 90 in the manuscript, we have described the diagnostics of measles and in the Discussion, regarding overlapping symptoms such as fever and rash amidst several pathogens, we considered the differential diagnoses.

“The diagnosis can be confirmed with molecular techniques like reverse transcription polymerase chain reaction or through serology, with measles virus-specific immunoglobulin (Ig)M antibody peaks seen when the rash is almost confluent, and the peak of IgG antibody observed with the resolution of symptoms, at which point the patient is no longer considered infectious”.

Similarly, for syphilis in lines 104 up to 110, we described briefly the techniques utilized frequently for the diagnosis of syphilis.

“Although spirochetes may be visualized under dark-field microscopy of specimens collected from the primary syphilitic chancre, in addition to the utilization of other, more sensitive molecular techniques such as nucleic acid amplification to detect T. pallidum genes. Nevertheless, serology remains the technique most commonly used in clinical practice for the diagnosis of secondary, latent, or tertiary syphilis, involving non-treponemal and treponemal tests; the former is utilized for screening and the latter for confirmatory diagnosis”.

>>As suggested, in the revised manuscript (Lines 218-221) we have rephrased the sentences to clarify the planned course of further workup and treatment. Further in the Discussion (Lines 328-330) we have described and added references to recent therapeutical recommendations for treating syphilis.

“(Lines 218-221) The patient was discharged from the hospital on day 7 after the onset of symptoms with clinical recovery from measles, and with a reactive RPR (titer of 1:16). A follow-up appointment with a repeated RPR to start treatment with benzathine benzylpenicillin G was advised.”

“(Lines 328-330) Penicillin remains the drug of choice to treat syphilis, including doxycycline has been described to be an acceptable alternative option to treat early and late latent syphilis, if penicillin cannot be used.”

>> As suggested, we have described a pragmatic and systemized approach to a patient presenting with fever, and rash. In addition, this manuscript also describes a situation and a possibility when a mono-infection might obscure other occult infections. We hope to heighten the clinical judgment of clinicians working in primary care as to the one the case had presented. In the Case Report, we described the meticulous physical examination and identification of the rash in Figure 2, which prompted requesting serological and confirmatory tests. Further, serology for STDs is not routinely performed in clinical practice unless otherwise indicated. As suggested we have added the following sentence with the RPR titer demonstrating the latency and the planned further workup and treatment.

“The patient was discharged from hospital on day 7 after the onset of symptoms with clinical recovery from measles, and with a reactive RPR (titer of 1:16). A follow-up appointment with a repeated RPR to start treatment with benzathine benzylpenicillin G was advised.”

>> In the Discussion, we categorically excluded etiologies considered in the differential diagnoses based on the available laboratory data. We direct the attention to lines 235 to 247, where we discuss potential blood-borne infections like HIV and Hepatitis B virus. Further, in the third and fourth paragraphs of the Discussion, we describe common viruses and caution on careful interpretation of serology results. Lastly, we discuss some bacterial etiologies that were considered in the differential diagnoses.

“(Lines 235-247) When formulating differential diagnoses for the presented case, the social lifestyle that posed several risks of pathogen transmission needs to be considered. An important one would be the potential exposure to HIV, as acute HIV seroconversion may exhibit similar clinical manifestations to the present case, regardless of the known date of last exposure, in view of the prolonged incubation period [46, 47]. Relatedly, antibodies against other blood-borne viruses, such as hepatitis B virus, were detected. In Thailand, vaccination against the hepatitis B virus was included in the nationwide expansion of the immunization program in 1992 [48]. We therefore postulate that the patient in the present case may not have received vaccination against hepatitis B virus. The detection of existing antibodies may be from an acquired infection in the past through exposure from tattooing or unsafe sexual practices with multiple partners. However, we could not further demonstrate any hepatitis B core antibody to reflect on the disease state during this admission.”

“(Lines 248-260) A vaccine in the form of a live attenuated combination of measles, mumps and rubella was introduced in Thailand under the expanded immunization program in 2008. Based on this information, we considered the patient in this case may have been naïve to these viruses. Hence, other viral etiologies such as rubella and parvovirus B19 infections may also manifest with similar clinical findings to the presented case [49, 50]. However, the classical manifestation of rubella in adults includes arthralgias, similar to the symptoms caused by arthritogenic viral infections [44, 51-53]. Similarly, parvovirus B19 infection in adults may manifest with arthralgia, including enanthemic eruptions of Koplik’s spots [54]. These prodromal enanthemic eruptions are considered pathognomonic to measles, but may occur with other viral infections. Koplik’s spots were present in 717 of 3023 cases of suspected measles [55]. While a third of the cases in the same cohort had Koplik’s spots from measles, less than a quarter were caused by rubella, and a little over 5% were caused by other viruses.”

 “(Lines 261-272) Further, a previous study demonstrated that serological assays for rubella performed less accurately in the sense that false-positive findings resulted from cross-reaction with other viral infections or were associated with underperformance with increased circulating serum proteins like rheumatoid factor [56, 57]. Hence, we advocate clinicians to interpret serology with caution. Furthermore, a previous report observed seroconversion in rubella 3–4 days after rash onset [58]. Retrospectively, in the present case, serum samples for serologic investigations were collected on the third day of illness, representing the second day after the onset of rash. We therefore considered that the serological results for rubella may have been false positives in the present case. Additionally, the epidemiology of measles in Thailand is far more common compared to rubella. Furthermore, other hyper-inflammatory states similar to Kawasaki disease may also show similar manifestations to the present case [59].

>> As suggested, we have revised the flow by moving Lines, 309-329 to 261 to 281, in the revised manuscript,

>> Unfortunately, the quality of the photograph of the Koplik’s spots was of poor quality and not pristine enough to be included in this manuscript. We also regret this a lot, as it would make the manuscript more robust.

>> We considered the rephrase the title for more reader clarity of what is described in this manuscript. In the revised manuscript, the title presently is as follows.

“A case report of secondary syphilis co-infected with measles: a diagnostic dilemma with fever and rash”.

>> We have also revised Line 29-30 as follows.

"Since viruses are more frequently associated with fever and rash, these symptoms are thus impetuously termed viral exanthem."

>> Further, as suggested, we have simplified Line 1 to A case report of secondary syphilis co-infected with measles: a diagnostic dilemma with fever and rash, and we have rephrased the sentence for simplification (Lines 101-104)

“The disease left untreated results in the seeding of bacterium (spirochete) within the body during the secondary stage, also known as “roseola syphilitica” and the cutaneous manifestation appearing during this stage may mimic other diseases.”

>> In the revised manuscript, we have deleted the words notorious and neighborhood.

>> In preparation of this manuscript, we took great caution in avoiding using terms that might attribute to stigmatizing as recommended by Marcus & Snowden, STD Jan 2020.

We have rephrased the sentence for (Lines 329-330) as follows.

"Nonetheless, the patient had a substantial history of increased risk of exposure to sexually transmitted diseases on multiple occasions"

Round 2

Reviewer 2 Report

N/A